# Repression of *ctrA* and *chpT* by a transcriptional regulator of the Xre family that is expressed by RpoN3 and its cognate activator protein in *Cereibacter sphaeroides*

**Benjamín Vega-Baray, José Hernández-Valle, Sebastián Poggio, Laura Camarena** [ID]*

Departamento de Biología Molecular y Biotecnología, Instituto de Investigaciones Biomédicas, Universidad Nacional Autónoma de México, Mexico City, Mexico

* rosal@unam.mx

## Abstract

*Cereibacter sphaeroides* is an α-proteobacteria that has two flagellar systems. Fla1 directs the assembly of a single subpolar flagellum, and Fla2 directs the assembly of multiple polar flagella. The *fla2* genes are controlled by the two-component system CckA/ChpT/CtrA. In the wild-type strain, the *fla2* genes are not expressed under the growth conditions commonly used in the laboratory, and thus far, their expression has only been reported in strains carrying either a gain-of-function version of CckA or a null mutation in *osp*, a negative regulator of CckA. In this work, the differential swimming response of two Fla2+ strains in response to the inclusion of a divalent ion in the culture medium was investigated. This analysis led to identifying a new transcriptional regulator of the XRE family, XrpA. This protein severely reduces the expression of *ctrA* and, consequently, the expression of the genes activated by this transcription factor. We show that XrpA binds to the control region of *ctrA* and *chpT,* suggesting that XrpA directly represses their expression. Additionally, we determined that RpoN3, one of the four RpoN paralogues of RpoN present in *C. sphaeroides*, and its cognate activator protein AprX are required for the expression of *xrpA*. XrpA is conserved in several species of Rhodobacterales and a σ54 promoter consensus sequence is present in its control region and a homologue of AprX cooccurs with it. These results support the idea that these proteins form a novel regulatory module that controls the TCS CckA/ChpT/CtrA in *C. sphaeroides* and other related species.

## Introduction

*Cereibacter sphaeroides* is a purple nonsulfur α-proteobacteria with versatile metabolism that is able to grow photoheterotrophically, photoautotrophically, or heterotrophically [1]. This species was reported to be motile [2], and later reports indicated that this microorganism swims via a single flagellum localized subpolarly [3]. Several studies have identified the components required for the assembly of this structure, as well as the regulatory system controlling the expression of the flagellar genes [4]. Like in other bacterial species,

**Data availability statement:** All relevant data are within the manuscript and its Supporting Information files.

**Funding:** This work was partially supported by Dirección General de Asuntos del Personal Académico- Programa de Apoyo a Proyectos de Investigación e Innovación Tecnológica (DGAPA-PAPIIT), de la Universidad Nacional Autónoma de México, grant IN215023 and by Consejo Nacional de Humanidades Ciencia y Tecnología (CONAHCYT), grant CBF2023-2024-501 (to LC). BVB was supported by postdoctoral fellowship 703928 from CONAHCYT. The funders had no role in study design, data collection and analysis, decision to publish, or preparation of the manuscript.

**Competing interests:** No authors have competing interests.

the flagellar genes are expressed following a transcriptional hierarchy of four tiers. At the top of the hierarchy, the master regulator protein FleQ together with the alternative sigma factor RpoN2 ($\sigma^{54}$-2) activate the expression of the genes in the hierarchy [5,6]. It is known that RpoN alone is unable to initiate transcription until an activator protein of the bacterial enhancer binding protein (bEBP) family stimulates open complex formation through the hydrolysis of ATP [7–9], FleQ belongs to the EBP family of proteins. Interestingly, in contrast to other bacterial species, which commonly have only one *rpoN* gene and several bEBPs, *C. sphaeroides* has 4 different *rpoN* genes and 5 bona fide bEBPs [5]. It has been demonstrated that RpoN1 and RpoN2 are both specific for the transcription of a particular set of genes, *i.e.,* RpoN1 is required to express genes involved in nitrogen fixation (*nif*), and RpoN2 is specific for the expression of flagellar genes [5]. It was reported that the identity of the -11-promoter position is relevant for the differential recognition of these promoters [10]. Additionally, the bEBPs required to express the *nif* and *fla* genes are specific for activating a particular RpoN protein, *i.e.,* the bEBP NifA activates RpoN1-mediated transcription, and FleQ and the heterocomplex FleQ/FleT activate RpoN2; The EBP FleT is particular because it lacks the DNA binding domain and, by itself, does not activate the flagellar $\sigma^{54}$-dependent promoters. The expression of *fleT* depends on FleQ and RpoN2, and once synthesized, it forms hetero-oligomeric complexes with FleQ. This interaction is relevant for controlling the transition between the different classes of the flagellar hierarchy, as FleQ recognizes and activates the class II flagellar promoters, while the FleQ/FleT complex recognizes the class III promoters. [6,11]. To date, no information is available for RpoN3 or RpoN4 or for the two other bEBPs present in the *C. sphaeroides* genome, except that they are not involved in the expression of the *nif* or *fla* genes.

After the early characterization of the flagellar system of *C. sphaeroides*, the genome sequence of this bacterium revealed the presence of another set of flagellar genes, named *fla2,* to distinguish it from previously characterized genes (*fla1*) [12]. At that time, there was no evidence regarding the functionality of the *fla2* genes or evidence of their expression. However, a few years later, it was reported that the expression of the *fla2* genes results in the assembly of several polar flagella [13]. The expression of these genes was achieved by selecting swimming cells that were spontaneous suppressors of a Fla1- mutant strain [13]. This finding was significant because, in contrast to other species where a secondary flagellar system enables swarming across surfaces, in *C. sphaeroides*, the Fla2 flagella is dedicated to swimming [4]. In these mutants, the Fla2 + phenotype was caused by a gain-of-function mutation in the hybrid histidine kinase CckA (CckA$_{L391F}$), and in agreement with this finding, it was shown that the two-component system (TCS) formed by CckA, the phosphotransfer protein ChpT and the response regulator CtrA, is required to express the *fla2* genes [14]. Recently, it was reported that inactivation of the *osp* gene, which encodes a negative regulator of CckA, also results in a Fla2 + phenotype [15].

Phylogenetic studies suggested that the Fla1 system was acquired by *C. sphaeroides* from an ancestral gamma-proteobacteria, whereas the *fla2* system represents vertically inherited genes that are distinctive of the α-proteobacteria. Despite this, in the wild-type strain, the TCS CckA/ChpT/CtrA is inactive, and consequently, the *fla2* genes are not expressed under the growth conditions commonly used in the laboratory, where the constitutive synthesis and assembly of the Fla1 flagellum occurs [13,16]. To date, no conditions that promote sustained activation of the TCS CckA/ChpT/CtrA in the wild-type strain have been reported. However, irrespective of the type of mutation that activates the TCS (*i.e.*, CckA$_{L391F}$ or Δ*osp*), it has been observed that the expression of the *fla2* genes is stimulated when the Fla2 + strains are grown photoheterotrophically with casamino acids as the carbon source. In contrast, their expression is low when 15 mM succinic acid is used as the carbon source [14,15].

The CckA/CtrA/ChpT system is conserved in many α-proteobacteria and in some of them, genes related to cell division and cytokinesis are under its control making it essential [17,18]. In *Caulobacter crescentus*, where the system is essential, different mechanisms adjust the expression, activity, localization, and stability of the three components of the system in response to internal and external cues. These cues are sensed by various regulatory proteins that directly or indirectly modify one or more of these processes [17–20]. In *C. sphaeroides*, the TCS CckA/ChpT/CtrA is not essential, and CtrA controls the expression of more than 300 genes, including those required for the synthesis and assembly of the Fla2 flagellum, gas vesicles, photosynthetic complexes, as well as genes involved in several stress responses [11,21].

Importantly, the mechanisms involved in the control of the CckA/CtrA/ChpT system vary depending on the species [17].

In this work, we report a new regulatory protein of the Xre family of transcription factors that represses the expression of *ctrA* and consequently reduces the expression of the *fla2* genes and other genes activated by this transcription factor. We also found that the expression of the gene encoding this novel regulatory protein is dependent on a specific paralog of RpoN, RpoN3, and its cognate bEBP, AprX.

## Materials and methods

### Strains, plasmids, oligonucleotides and growth conditions

Bacterial strains and plasmids used in this work are listed in Table 1. *Cereibacter sphaeroides* WS8N [22], and its derivatives AM1 [23], or BV11 [15] were used as parental strains. *C. sphaeroides* was grown at 30 °C in Sistrom's minimal medium that includes 15 mM succinic acid as a carbon source [24]. When specified, the carbon source was substituted by 0.2% casamino acids. As needed, the culture medium was supplemented with the concentrations of divalent cations indicated. Photoheterotrophic liquid cultures were grown under continuous illumination in completely filled screw-cap tubes. Heterotrophic liquid cultures were incubated in the dark with orbital shaking at 200 rpm. *E. coli* was grown in Luria-Bertani (LB) medium [25] at 37ºC. When required, antibiotics were added at the indicated concentrations: for *C. sphaeroides*, kanamycin (25 µg/ml), tetracycline (1 µg/ml), spectinomycin (50 µg/ml), and hygromycin (20 µg/ml for liquid cultures and 150 µg/ml for plates); for *E. coli*, kanamycin (50 µg/ml), spectinomycin (50 µg/ml), ampicillin (100 µg/ml), hygromycin (20 µg/ml for liquid cultures and 200 µg/ml for plates) and nalidixic acid (20 µg/ml).

### Oligonucleotides

The oligonucleotides used in this work were purchased from oligo T4 (Irapuato, Gto. Mex.) and are listed in S1 Table.

### Motility assays

Motility was tested in soft agar plates containing Sistrom's minimal medium, which includes 15 mM succinic acid or 0.2% casamino acids as a carbon source. Bacto agar was added at a final concentration of 0.2%. As needed, the culture medium was supplemented with the concentrations of divalent cations indicated. Plates were spotted with 2µl of a stationary-phase culture, and aerobically incubated in the dark at 30ºC. Swimming capacity was recorded as the ability of bacteria to move away from the inoculation point after incubation at 30 °C. Diameters of the swimming rings were measured by considering the outermost spots around the ring. If the inoculum slides on the surface of the culture medium, the resulting swimming ring is lopsided, and in this case, the shortest diameter of the ring was taking into account.

**Table 1. Strains and plasmids used in this work.**

| Strain or Plasmid | Description | Reference |
|---|---|---|
| *Cereibacter sphaeroides* strains | | |
| AM1 | SP13 derivative; *cckA*$_{L391F}$ | [23] |
| BV10 | SP20 derivative; Δ*osp*::Hyg | [15] |
| BV11 | SP13 derivative; Δ*osp*::Hyg | [15] |
| BV12 | AM1 derivative; Δ*osp*::Hyg | [15] |
| JHV3 | AM1 derivative; *mcpB*::*uidA-aadA* | [11] |
| JHV7 | AM1 derivative; *gvpN*::*uidA-aadA* | [21] |
| SS1 | AM1 derivative; *flhA2*::*uidA-aadA* | [13] |
| LC7 | AM1 derivative; Δ*ctrA*::Hyg | [11] |
| LC5 | SP13 derivative; *cckA*::Ω | [14] |
| SP13 | WS8N derivative; Δ*fleQ*::Kan | [6] |
| SP20 | WS8N derivative; Δ*fliF1*::*aadA* | Laboratory collection |
| BV20 | BV11 derivative; Δ*xrpA*::aadA | This work |
| BV21 | BV10 derivative; Δ*rpoN1*::Kan | This work |
| BV22 | BV10 derivative; Δ*rpoN2*::Kan | This work |
| BV23 | BV10 derivative; Δ*rpoN3*::Kan | This work |
| BV24 | BV10 derivative; Δ*rpoN4*::Kan | This work |
| BV25 | BV10 derivative; Δ*nifA*::Kan | This work |
| BV26 | BV10 derivative; ΔRSWS8N_04525::Kan | This work |
| BV27 | BV10 derivative; Δ*fleQ* | This work |
| BV28 | BV10 derivative; Δ*aprX*::Kan | This work |
| BV30 | SP13 derivative Δ*xrpA*::*aadA* | This work |
| WS8N | Wild type strain Nal$^R$ | [22] |
| *Escherichia coli* strains | | |
| TOP10 | Cloning strain | Invitrogen |
| Rosetta | Protein expression strain | Novagen |
| **Plasmids** | | |
| pMAL_cRI | Expression vector for MBP-tagged proteins, Ap$^R$ | New England Biolabs |
| pIJ963 | Plasmid source of the Hyg cassette | |
| pJQ200_Δ04525::Kan | pJQ200mp18 carrying Δ04525::Kan | [5] |
| pJQ200_ΔaprX::Kan | pJQ200mp18 carrying Δ*aprX*::Kan | This work |
| pJQ200_ΔnifA::Kan | pJQ200mp18 carrying Δ*nifA*::Kan | [10] |
| pJQ200_ΔrpoN1::Kan | pJQ200mp18 carrying Δ*rpoN1*::Kan | [5] |
| pJQ200_ΔrpoN2::Kan | pJQ200mp18 carrying Δ*rpoN2*::Kan | [5] |
| pJQ200_ΔrpoN3::Kan | pJQ200mp18 carrying Δ*rpoN3*::Kan | [5] |
| pJQ200_ΔrpoN4::Kan | pJQ200mp18 carrying Δ*rpoN4*::Kan | [5] |
| pJQ200_Δxrpa::aadA | pJQ200mp18 carrying Δ*xrpA*::*aadA* | This work |
| pJQ200mp18 | Suicide vector for *C. sphaeroides* | [26] |
| pMAL-xrpA | Expression vector for MBP-XrpA, Amp | This work |
| pRK_aprX | pRK415 expressing *aprX* | This work |
| pRK_rpoN1 (pRS104) | pRK415 expressing *rpoN1* | [5] |
| pRK_rpoN2 (pRS200) | pRK415 expressing *rpoN2* | [5] |
| pRK_rpoN3 (pRS206) | pRK415 expressing *rpoN3* | [5] |
| pRK_xrpA | pRK415 expressing *xrpA* | This work |
| pRK_xrpA::uidA | pRK415 carring the transcriptional fusion *xrpA*::*uidA* | This work |
| pRK415 | Expression vector used in *C. sphaeroides*, Tc$^R$ | [27] |

*(Continued)*

**Table 1.** (Continued)

| Strain or Plasmid | Description | Reference |
|---|---|---|
| pTZ18R_aprX | pTZ18R carrying *aprX* | This work |
| pTZ18R_ΔaprX::Kan | pTZ18R carrying Δ*aprX*::Kan | This work |
| pTZ18R_ΔxrpA::aadA | pTZ18R carrying Δ*xrpA*::*aadA* | This work |
| pTZ18R_ΔxrpA::uidA-aadA | pTZ18R carrying Δ*xrpA*::*uidA-aadA* | This work |
| pTZ18R_xrpA | pTZ18R carrying xrpA | This work |
| pTZ18R/19R | Cloning vectors, Ap^R | [28] |
| pTZ19R-BamHI⁻ | Cloning vectors, Ap^R | Laboratory collection |
| pUC4K | Plasmid source of the Kan cassette | Pharmacia |
| pWM5 | vector source of the *uidA-aadA* cassette | [29] |

## Mutant isolation and plasmids constructed in this work

All the mutants used in this work were obtained by gene replacement using the suicide plasmid pJQ200mp18 [26] carrying the appropriated mutant allele. To inactivate *xrpA*, the following constructions were obtained: the plasmid pTZ_xrpA was obtained by cloning a 1,267 bp PCR fragment obtained with the oligonucleotides Fw XREcro_Xba and Rv XREcro_Sac into pTZ18R previously digested with SmaI. The orientation of the cloned fragment in the resulting plasmid was selected so that a XbaI digestion released the insert from the plasmid. Plasmid pTZ_xrpA::aadA was obtained by cloning in the coding region of *xrpA* the spectinomycin resistance gene *aadA*. For this, pTZ_xrpA was digested with SalI, termini were repaired with T4 polymerase, and then ligated with a PCR product carrying the gene *aadA*, which encodes an aminoglycoside (3")(9) adenylyltransferase that confers spectinomycin resistance. The XbaI fragment carrying the mutant allele *xrpA*::*aadA* was subcloned into pJQ200mp18 previously digested with XbaI. To replace the *xrpA* wild-type gene, the suicide plasmid pJQ_xrpA::aadA was introduced to *C. sphaeroides* by conjugation and the double recombination events were selected as described previously [30,31]. The presence of the correct gene replacement was verified by PCR.

To obtain the strain BV28 (Δ*aprX*::Kan), a 2,544 bp PCR product obtained with the oligonucleotides ACT159_1 and ACT159_2, was cloned in pTZ19R_BamH1⁻ previously digested with XbaI. The resultant plasmid, pTZ_159 was digested with BamHI and MscI that delete an internal 817 bp fragment of *aprX*. The large fragment of this reaction was purified and ligated with the Kan^R cassette (1.2 kb) once their 5' ends were repaired. The Kan^R cassette was obtained by PCR using the plasmid pUC4K and the oligonucleotides Kanfw1_Eco and Kanrev_EcoM. The fragment carrying the mutant allele Δ*aprX*::Kan was obtained by digesting pTZ18R_ΔaprX::Kan with XbaI. This fragment was subsequently cloned into pJQ200mp18 previously digested with XbaI. The mutant strain was obtained following the procedure described above.

Plasmid pRK_xrpA was generated by cloning in pRK415 the 430 bp PCR product obtained with the oligonucleotides Fw_Xre_Xba and Rv_Xre_HindIII using genomic DNA in the reaction. Plasmid pRK_aprX was generated by cloning in pRK415 [27] the 2,544 bp PCR product obtained with the oligonucleotides ACT159_1 and ACT159_2. Plasmid pRK_xrpA::uidA (transcriptional fusion) was obtained by cloning the SmaI fragment carrying the *uidA-aadA* cassette into pTZ_xrpA previously digested with SalI and once their 5′-ends were repaired with T4 polymerase. The proper orientation of the *uidA-aadA* cassette was selected to obtain

the transcriptional fusion *xrpA*::*uidA*. The resultant plasmid was used as an amplification substrate using the oligonucleotides Fw_XREcro_xba and uidCOOH_Xba to obtain a 2,639 bp PCR product with the fusion *xrpA*::*uidA*. This product was then cloned in pRK415 previously digested with XbaI. The proper orientation was selected so that the expression of *xrpA* were in opposite direction of the *lac* promoter present in pRK415.

BV21, BV22, BV23 and BV24 were isolated using the pJQ200 plasmids carrying Δ*rpoN1*::Kan, Δ*rpoN2*::Kan, Δ*rpoN3*::Kan or Δ*rpoN4*::Kan alleles [5], and BV10 as recipient strain.

BV25, and BV26 were isolated using the pJQ200 plasmids carrying Δ*nifA*::Kan and Δ*04525*::Kan (Δ*187*::Kan, in reference [6]) alleles, and BV10 as recipient strain.

### β-glucuronidase activity assay

Cell-free extracts from exponential phase cultures were tested for β-glucuronidase activity following previously reported protocols [32]. Activities were calculated using a standard curve of 4-methyl-umbelliferone. Specific activities are expressed as pmol of 4-methyl-umbelliferone formed per min per mg of protein. Protein concentration was determined by the Bradford method (Bradford protein assay dye reagent; Bio-Rad, USA), using bovine serum albumin as standard.

### Electron microscopy

Cells were grown heterotrophically using Sistrom's minimal medium and supplemented as needed with the indicated concentration of $MgSO_4$. A few microliters from the cell suspension were placed on a carbon coated grid (Formvar carbon B on 200 mesh copper) and allowed to stand for 1 min at room temperature. Grids were negatively stained with 1.0% phospho-tungstic acid (pH 7.0) for 1 minute and rinsed with distilled water, dried, and observed using a JEM-1200EXII (Jeol) electron microscope. Grids were negatively stained with 1.0% uranyl acetate for 1 minute and rinsed with distilled water, dried, and observed using a JEM-1200EXII (Jeol) electron microscope.

### Antibodies and Western blotting

Polyclonal antibodies were raised in female BALB/c mice against MBP-XrpA following standard procedures described previously [33]. For immunoblotting, proteins were separated by SDS–PAGE, blotted onto nitrocellulose and tested with the primary antibody, after blocking in 2% (w/v) nonfat milk–phosphate-buffered saline (PBS). Anti-mouse IgG-AP conjugate (Sigma-Aldrich) at a 1/30000 dilution was used as the secondary antibody [33]. Detection was carried out with CDP-Star substrate (Applied Bio-systems) and visualized using a CDD camara. Gel electrophoresis for detection of the FlaA and CtrA proteins was performed using 12% PAGE, while XrpA detection was performed on 15% PAGE.

### MBP-XrpA overexpression and purification

To overexpress XrpA, the coding region of *xrpA* was amplified using the oligonucleotides FW_xre_EcoRI and Rv Xre HindIII. After gel-purification, the 366 bp product was cloned into pMAL-cRI. MBP-XrpA was purified from a culture of the Rosetta/pMal_xrpA strain that was induced for 4 h with 0.1 mM IPTG. Cells were harvested and resuspended in buffer containing 50 mM Tris (pH 8), 50 mM NaCl, 5% (v/v) glycerol and 1 mg/ml lysozyme. Cell lysis, protein binding to the amylose resin, washing and elution were carried out following a protocol reported elsewhere [34].

## Gel mobility shift assays

The DNA fragments that include the regulatory region of *ctrA*, *chpT* and *cckA* were obtained by PCR using the oligonucleotides FW_ctrAp and Rv_ctrAp for *ctrA*, chpT_mut_up2 and chpT_mut_up1 for *chpT*, and CckAprom and RvcckaTM_Pst for *cckA*. The PCR products of 396, 603 and 327 bp, respectively, were purified using QIAquick gel extraction kit (QIAGEN) and quantified by $OD_{260}$. Binding reactions were carried out in 1X EMSA binding buffer (10 mM Tris-HCl pH 7.5, 100 mM KCl, 1mM EDTA, 0.1mM DTT, 5% (v/v) glycerol, 0.01% BSA) in a total volume of 10 µl [35]. Each reaction included 50 ng of DNA from the indicated regulatory region and 100 ng of XrpA. After 30 min of incubation at room temperature, the samples were loaded on a 4% native polyacrylamide gel and run at 100 V in buffer containing 40 mM Tris-acetate pH 7.8, 2.5mM EDTA. After electrophoresis, DNA was detected by ethidium bromide staining.

## Bioinformatic analysis of the sequences

For each genome in Fig 11, the region upstream of *xrpA* was obtained from the NCBI database (https://www.ncbi.nlm.nih.gov/). These sequences were searched for the presence of DNA motifs using MEME version 5.4.1 [36]. The secondary structure of XrpA was predicted using Psipred [37]. The 3D model of XrpA was obtained from AlphaFold2 [38,39]. Synteny was evaluated using the Genomic Context Visualizer (GeCoViz) [40].

## Phylogenetic analysis

The Rhodobacterales species were selected if their genome was complete or > 95%. The RpoC protein was identified by BLASTP [41]. The RpoC proteins were aligned with MUSCLE [42]. The phylogenetic tree was constructed by neighbor joining method [43].

# Results

## Differential swimming of strains AM1 and BV11 when $MgSO_4$ is included in the culture medium

As reported previously, strains showing a Fla2 + phenotype were the result from either a gain-of-function mutation in CckA (Group 1) or from the loss of the negative regulator of CckA, Osp (Group 2). The parental strain used to obtain these mutant strains was SP13, which displays a Fla1⁻Fla2⁻ phenotype due to a mutation in the master regulator of the *fla1* genes (*fleQ*), and the inactivity of the CckA/ChpT/CtrA TCS. Strain AM1 represents the first group of Fla2 + mutants that express the gain-of-function version of CckA, $CckA_{L391F}$ [23]. Strain BV11 represents the second group of Fla2 + mutants, which carry the Δ*osp* allele [15]. As shown in Fig 1, motility proficiency is similar between the AM1 and BV11 strains under conditions that stimulate the expression of the *fla2* genes (photoheterotrophic growth in 0.2% casamino acids) or under conditions that reduce it (heterotrophic growth in 15 mM succinic acid) (Fig 1 and S1 Fig).

We unexpectedly discovered that, under heterotrophic growth conditions, the addition of MgSO₄ to the culture medium significantly enhanced the swimming of AM1 when 15 mM succinic acid was used as the carbon source, resulting in an approximately 4.5-fold increase in the expansion of the swimming ring. In contrast, BV11 exhibited only a modest improvement of approximately 2-fold (Fig 2A and 2B). In plates incubated under photoheterotrophic conditions, the inclusion of MgSO₄ similarly promoted the swimming of both strain (~2.5-fold) (Fig 2C and 2D). These results suggest that the factors responsible for the differences between these strains do not play a significant role under photoheterotrophic conditions. Therefore, we

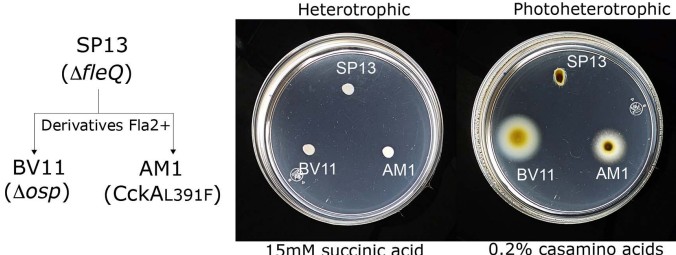

**Fig 1. Swimming proficiency of strains SP13, AM1 and BV11 grown under different conditions.** Soft-agar plates containing the indicated carbon source were inoculated with the indicated strain and incubated either under heterotrophic or photoheterotrophic conditions for 48 h. At the left the relevant mutations of each strain are shown, the diagram also shows that SP13 is the parental strain of AM1 and BV11.

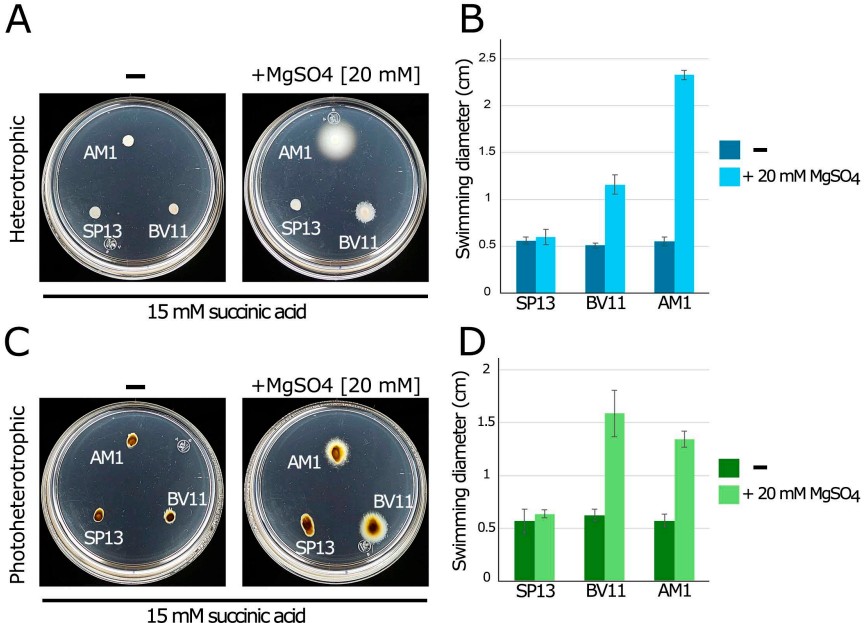

**Fig 2. Swimming proficiency of AM1 and BV11 in the presence or absence of MgSO$_4$.** Soft-agar plates containing Sistrom's minimal medium containing 15 mM succinic acid as a carbon source without MgSO$_4$ (-) or supplemented with 20 mM of MgSO$_4$ (panels A and C) were inoculated with the indicated strains. Plates were incubated under heterotrophic or photoheterotrophic conditions for 48 h. SP13 was included as a negative control. The right side of the figure (panels B and D) shows the average diameter of the swimming ring of each strain. Measurements were taken from at least three independent experiments. The values are the average of three independent experiments, standard deviations are shown.

focused our attention on analyzing the stimulatory effect of MgSO$_4$ on the swimming behavior of strain AM1 when grown heterotrophically in 15 mM succinic acid as the carbon source.

Under these growth conditions, it was found that other divalent cations also stimulate the swimming of AM1 cells, whereas KCl and NaCl do not (S2 Fig). In these experiments, only low concentrations of MnCl$_2$ and CaCl$_2$ could be tested because higher concentrations of these salts resulted in the formation of precipitates. Therefore, in subsequent experiments MgSO$_4$ was used since no precipitation of the culture medium was observed.

## Effect of MgSO$_4$ on flagellar formation in the AM1 strain

To determine whether the inclusion of MgSO$_4$ improved the swimming ability of AM1 cells by increasing the number of flagella per cell, we carried out electron microscopy (EM) observations of AM1 cells grown heterotrophically in 15 mM succinic acid with 0, 2 and 20 mM MgSO$_4$ added to the culture medium. As shown in Fig 3A, no flagellated cells were detected in the absence of MgSO$_4$, but their presence increased appreciably when MgSO$_4$ was included in the culture medium. To further support this observation, we analyzed the amount of flagellin (FlaA) in total AM1 cell extracts, which reflects flagellar synthesis. This analysis revealed a conspicuous increase in flagellin levels, which increased in parallel with the concentration of

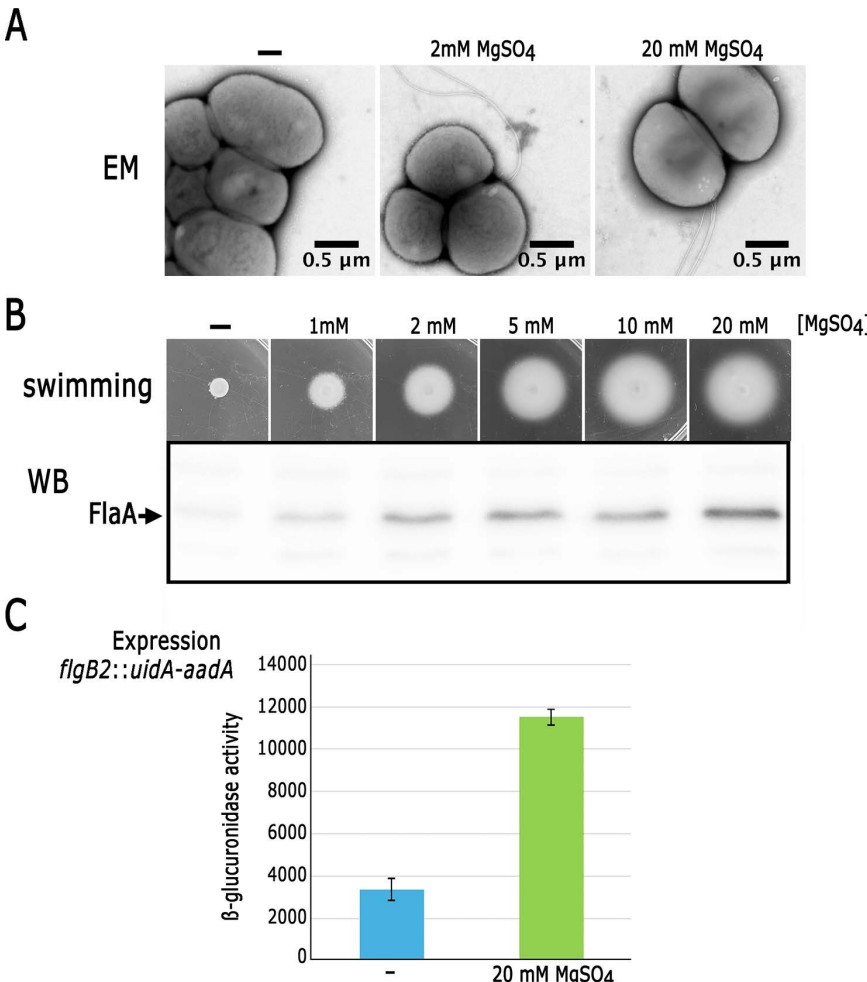

**Fig 3. Effect of MgSO$_4$ on the expression, synthesis, and assembly of the Fla2 flagella in strain AM1.** A, Electron microscopy of AM1 cells grown under heterotrophic conditions in Sistrom's minimal medium, either without MgSO$_4$ (-) or supplemented with 2 or 20 mM MgSO$_4$. B, AM1 cells inoculated on soft-agar plates containing 15 mM succinic acid and different concentrations of MgSO$_4$. The plates were incubated under heterotrophic conditions for 48 h. Below, flagellin detection (FlaA) by Western blot in total cell extracts from AM1 grown heterotrophically in liquid cultures containing 15 mM succinic acid with the indicated MgSO$_4$ concentration. C, β-glucuronidase activities from the transcriptional fusion *flgB2::uidA-aadA*. Strain AM1/pRK_flgB2::uidA-aadA was grown heterotrophically in Sistrom's minimal medium, either without (-) or supplemented with 20 mM MgSO$_4$. Activities are expressed as picomoles of methylumbelliferone formed per minute per milligram of protein. The values are the average of three independent experiments, standard deviations are shown.

MgSO₄ included in the culture medium, explaining the improved swimming proficiency of AM1 in response to MgSO₄ (Fig 3B). Furthermore, using the transcriptional reporter gene *uidA*, which encodes β-glucuronidase, we determined that MgSO₄ also stimulates the expression of the flagellar *flgB2* gene. FlgB2 is involved in the early assembly of the flagellar structure (Fig 3C). These results support the idea that MgSO₄ stimulates the general expression of *fla2* genes.

## A transcription factor of the XRE family counteracts the positive effect of MgSO₄ on swimming

To test if the activation of the *fla2* system in the AM1 strain in response to the presence of MgSO₄ in the culture medium was dependent on the $cckA_{L391F}$ allele, the wild-type allele of *cckA* was removed from the AM1 parental strain SP13, and the resulting strain (LC5) was complemented with a plasmid expressing $CckA_{L391F}$. The inclusion of MgSO₄ in the swimming plates containing 15 mM succinic acid in the culture medium did not promote swimming of this strain (S3 Fig). Another possibility was that deleting *osp* prevented the BV11 strain from responding to the presence of MgSO₄. To test this possibility, we introduced the Δ*osp* allele into AM1, and the motility of the resulting strain (BV12) was still stimulated by MgSO₄ (S3 Fig). These results indicate that the Δ*osp* allele does not prevent the stimulatory effect of MgSO₄, nor $CckA_{L391F}$ is responsible for this effect. Therefore, an unknown mutation must be responsible for this differential phenotype. The genome sequence of these strains has been previously reported [14,15], and their comparison revealed that AM1 has an additional mutation that is not present in BV11. This mutation is a substitution of a C for a T localized 46 bp upstream of the translation start site of RSWS8N_14710, which encodes a transcriptional regulator of the xenobiotic response element (XRE) family, that we named XrpA (XRE regulatory protein) (Fig 4A). A close inspection of the upstream region of *xrpA* revealed a sequence that conforms well with the consensus promoter sequence recognized by the alternative sigma factor RpoN or sigma-54 (σ⁵⁴) (Fig 4B) [44,45]. This type of promoters contain

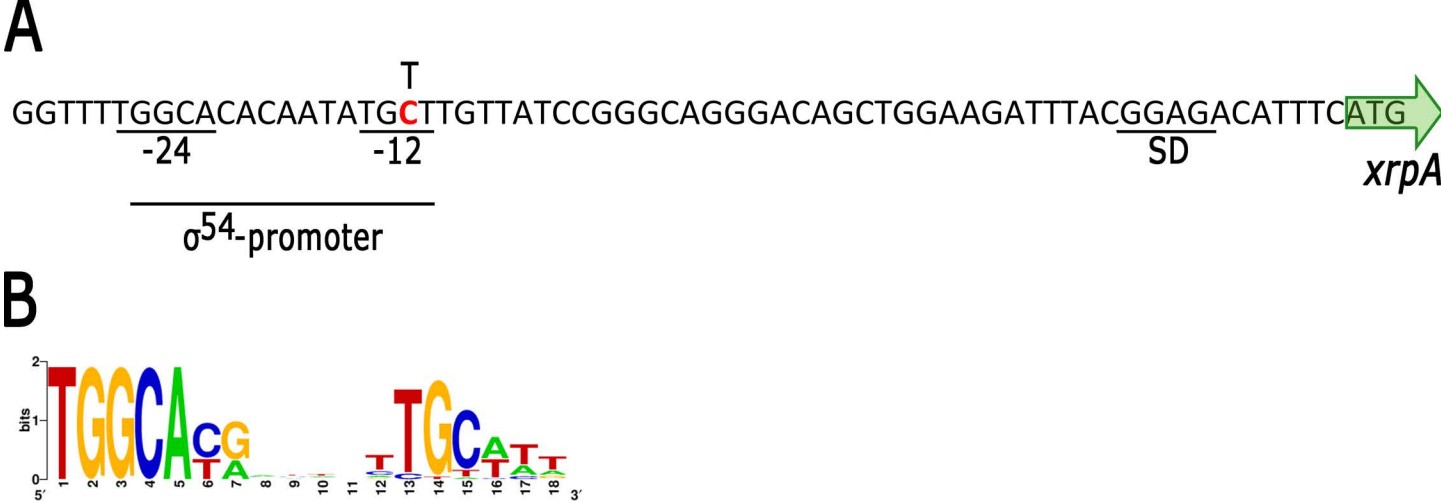

**Fig 4. Depiction of the regulatory region of RSWS8N_14710 and the mutation found in AM1.** A, the image shows the nucleotide sequence of the regulatory region of RSWS8N_14710; the start codon is indicated by a green arrow, and the Shine-Dalgarno (SD) sequence is underlined. The sequence resembling the σ⁵⁴ consensus promoter is underlined, the conserved -24 and -12 regions containing the GG and GC dinucleotides are also indicated. The substitution found in AM1 is shown above the sequence. B, Sequence LOGO representation of 27 σ⁵⁴-promoters previously reported for *S. typhimurium* [45], the LOGO was generated using the server http://weblogo.berkeley.edu/ [46].

conserved sequence elements located 24 and 12 bp upstream of the transcription start site. The dinucleotides GG and GC are highly conserved in each of these elements (Fig 4B). In AM1, the substitution of C for T changes the highly conserved dinucleotide GC at the -12 element (Fig 4A), suggesting that this change could reduce the expression of *xrpA* in AM1 and cause the differential response of strains AM1 and BV11 to MgSO$_4$.

To test this possibility, we deleted *xrpA* in BV11 to obtain the BV20 strain. In contrast to its parental strain, BV20 was able to assemble flagella and form a large swimming ring in 15 mM succinic acid/20 mM MgSO$_4$, like that produced by AM1 (Fig 5A and 5C). Importantly, like AM1, BV20 presented a reduced swimming ring in the absence of MgSO$_4$, indicating that the general repressive effect caused by 15 mM succinic acid was not affected by the absence of XrpA (Fig 5A). Additionally, BV20 was successfully complemented when *xrpA* was expressed from the pRK415 plasmid, indicating that no other undetected mutation was responsible for the observed phenotype (Fig 5B). It is also significant that deleting *xrpA* in SP13 was not enough to promote swimming consistent with the idea that activation of the TCS CckA/ChpT/CtrA is an indispensable prerequisite for the Fla2 + phenotype (S4 Fig).

These results indicate that XrpA prevents the stimulatory effect mediated by MgSO$_4$ and support the hypothesis that *xrpA* could be expressed from a σ$^{54}$-dependent promoter, which is mutated in the AM1 strain.

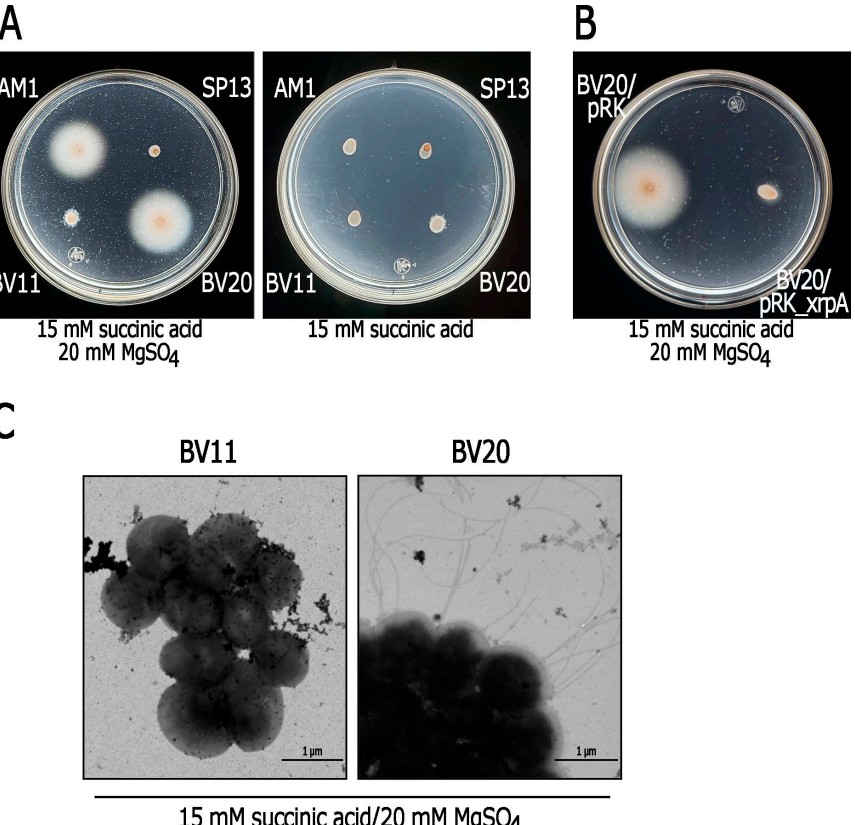

**Fig 5. Swimming proficiency and flagella formation upon deletion of *xrpA* in the BV11 strain.** A and B, Swimming plates prepared with minimal medium containing 15 mM succinic acid and 20 mM MgSO$_4$, or just 15 mM succinic acid were inoculated with the indicated strains and incubated heterotrophically for 48 h. C, Electron micrographs of BV11 (Δ*osp*) and BV20 (Δ*osp* Δ*xrpA*). Cells were grown heterotrophically in liquid medium containing 15 mM succinic acid/20 mM MgSO$_4$.

Consistent with a defect in the *xrpA* promoter, a reduced level of XrpA in AM1, compared with that observed in BV11, was detected by immunoblotting (Fig 6A). Restitution of *xrpA* in AM1 (using pRK_xrpA, which transcribes *xrpA* from the *plac* promoter), caused a significant reduction in its swimming proficiency (Fig 6C), which was due to its impaired ability to express FlaA and assemble flagella (Fig 6B and 6D). This phenotype was similar to that observed in BV11 when grown heterotrophically in minimal medium containing 15 mM

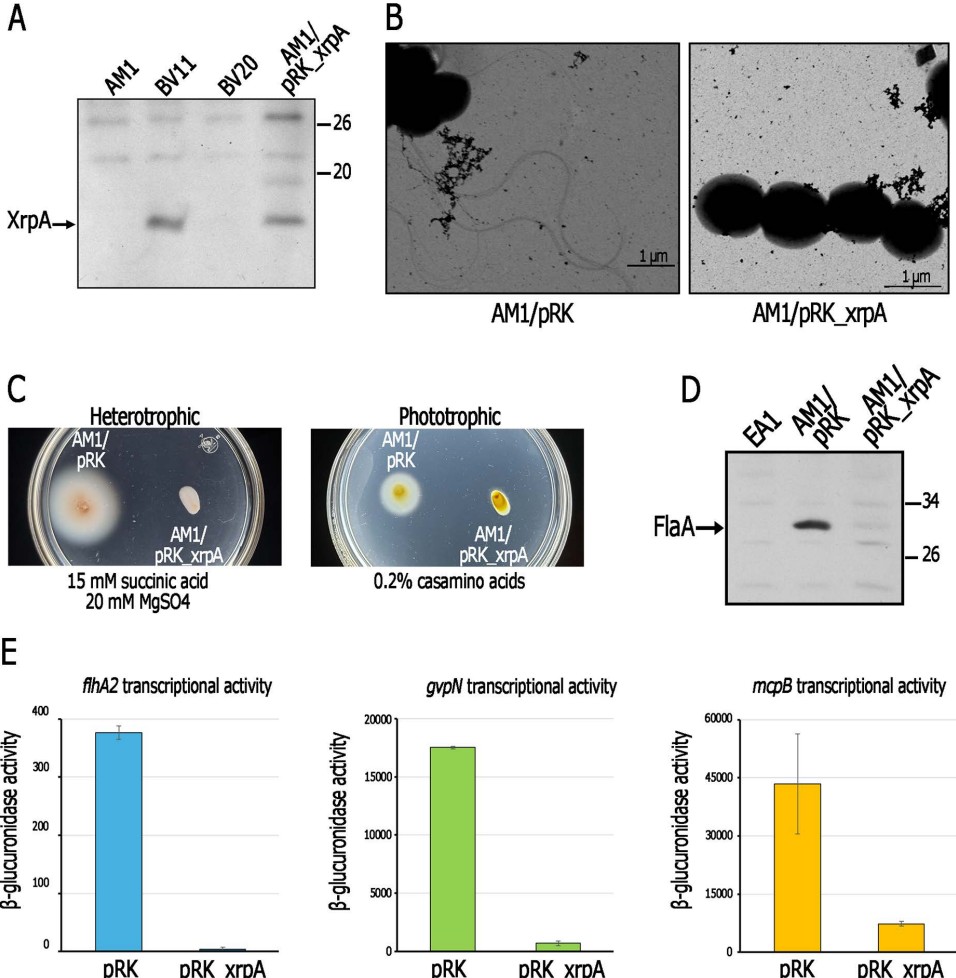

**Fig 6. Presence of XrpA in strains AM1 and BV11, and effect of pRK_xrpA on the expression of CtrA-dependent genes and flagella formation in AM1.** A, Immunodetection of XrpA in total cell extracts of AM1, BV11 and AM1/pRK_xrpA. Strain BV20 ($\Delta xrpA$) was included as a negative control. Cells were grown heterotrophically in Sistrom's minimal medium containing 15 mM succinic acid/20 mM MgSO$_4$. B, EM of strains AM1/pRK415 (AM1/pRK) and AM1/pRK_xrpA grown heterotrophically in liquid medium containing 15 mM succinic acid/20 mM MgSO$_4$. C, Soft-agar plates containing 1 μg/ml tetracycline and the indicated carbon source in the culture medium were inoculated with AM1/pRK415 (AM1/pRK) or AM1/pRK_xrpA. Plates were incubated under heterotrophic or photoheterotrophic conditions during 48 h. D, Immunodetection of flagellin (FlaA) in total cell extracts of AM1/pRK415 (AM1/pRK) and AM1/pRK_xrpA. Cells were grown heterotrophically in minimal medium containing 15 mM succinic acid/20 mM MgSO$_4$. E, β-glucuronidase activities of the chromosomal transcriptional fusions *flhA2::uidA-aadA*, *gvp*N::*uidA-aadA*, and *mcp*B::*uidA-aadA*, in AM1 derivative strains carrying either pRK415 (pRK) or pRK_xrpA. Total cell extracts of these strains were obtained from cultures grown heterotrophically in minimal medium containing 15 mM succinic acid/20 mM MgSO$_4$. Activities are expressed as picomoles of methylumbelliferone formed per minute per milligram of protein. The values are the average of three independent experiments, standard deviations are shown.

succinic acid/20 mM MgSO$_4$. Unexpectedly, the swimming of AM1/pRK_xrpA was also reduced when grown photoheterotrophically in minimal medium containing 0.2% casamino acids (Fig 6C), suggesting that the unregulated expression of XrpA leads to the inhibition of Fla2-dependent motility even under these conditions, in which BV11 (*xrpA+*) is able to swim (see Fig 1). These findings suggest that a mechanism that reduces the level of XrpA under certain growth conditions may exist. In line with this possibility, BV11 grown photoheterotrophically in 0.2% casamino acids showed a reduction in XrpA levels compared to those detected when the strain was grown heterotrophically in 15 mM succinic acid (S5 Fig). In addition, XrpA levels in AM1 were restored when XrpA was expressed from pRK_xrpA, regardless of the growth conditions (S5 Fig).

To investigate whether XrpA negatively affects the expression of CtrA-regulated genes or specifically downregulates *fla2*, we tested the expression of other genes known to be controlled by CtrA, such as *gvpN* (involved in gas vesicle formation) and *mcpB* (encoding a chemotactic receptor) [11,21]. For this purpose, we measured the amount of β-glucuronidase expressed from the chromosomal transcriptional fusions *gvpN::uidA-aadA*, *mcpB::uidA-aadA*, and as a control *flhA2::uidA-aadA* in AM1 derivative strains carrying either pRK415 or pRK_xrpA. These experiments revealed that, in all cases, the presence of XrpA significantly reduced the expression of the reporter gene (Fig 6E).

## XrpA controls the expression of *ctrA* and *chpT* and binds to their control region

XrpA belongs to the XreR family of transcriptional regulators. This family is one of the most common families in bacteria. Many proteins of this family have additional domains that bind small effector molecules or interact with other proteins to modulate their activity in response to environmental signals [47–49]. XrpA has 120 residues, its N-terminal region (residues 10 to 67) shows a HTH DNA-binding domain similar to the Cro/CI-HTH domain (COG1396); the C-terminal region (53 residues) is predicted to fold as a double alpha-helix that does not conform to any known conserved domain (S6 Fig). Presumably, like other Xre regulators, XrpA could bind DNA and repress the expression of its target genes.

Since XrpA negatively affects the expression of genes controlled by CtrA, it was hypothesized that XrpA could directly repress the expression of one or more components of the TCS CckA/ChpT/CtrA. To test this possibility, we examined whether XrpA could bind to the regulatory regions of *cckA*, *chpT* or *ctrA* through an electrophoretic mobility shift assay (EMSA). For these experiments, the upstream regions of these genes were obtained by PCR and incubated with the purified MBP-XrpA protein. As shown in Fig 7A, DNA fragments carrying the regulatory regions of *ctrA* and *chpT* showed a shift in electrophoretic mobility in the presence of MBP-XreA, whereas no shift was observed with the regulatory region of *cckA*. These findings suggest that XrpA may directly control the expression of *chpT* and *ctrA*. Furthermore, in AM1 background, the transcriptional fusions *ctrA::uidA-aadA* and *chpT::uidA-aadA* strongly reduced the expression of the reporter gene when XrpA was expressed from pRK415 (Fig 7B). As expected from these results, CtrA was barely detected in total cell extracts obtained from AM1/pRK_xrpA (Fig 7C).

## The expression of *xrpA* depends on RpoN3 and its cognate activator protein AprX

Our results suggest that in strain AM1, the reduced expression of *xrpA* is due to a mutation in its regulatory region that affects a putative σ$^{54}$-dependent promoter. To investigate the role of RpoN in controlling *xrpA* expression, we tested if the deletion of any of the *rpoN* genes of

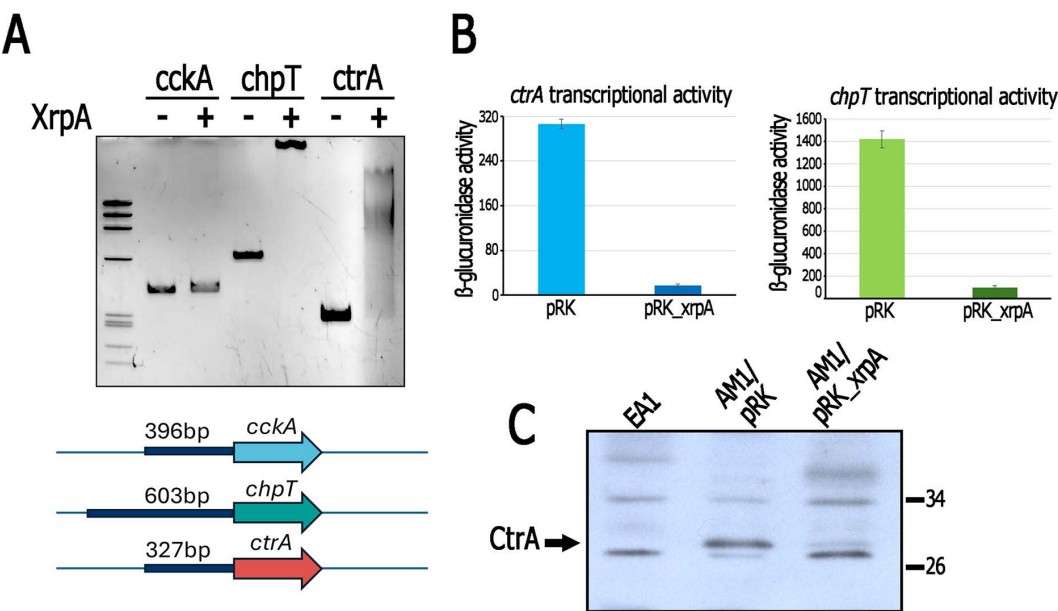

**Fig 7. XrpA binds to the regulatory regions of *chpT* and *ctrA* and negatively affects their expression.** A, the regulatory regions of *cckA*, *chpT* and *ctrA* (indicated by the dark blue box in the lower part of the panel) were incubated with MBP-XrpA (+) or supplemented only with binding buffer (-). The reactions were analyzed by polyacrylamide gel electrophoresis, and the DNA was visualized by ethidium bromide staining. B, β-glucuronidase activities promoted by the transcriptional fusions *ctr*A::*uidA* and *chp*T::*uidA* in AM1 derivative strains carrying either pRK415 (pRK) or pRK_xrpA. Total cell extracts were obtained from cultures grown heterotrophically in minimal medium containing 15 mM succinic acid/20 mM MgSO$_4$. Activities are expressed as picomoles of methylumbelliferone formed per minute per milligram of protein. The values are the average of three independent experiments, standard deviations are shown. C, Detection of CtrA by immunoblotting in total cell extracts of AM1 cells carrying pRK415 (pRK) or pRK_xrpA. Cells were grown heterotrophically in minimal medium containing 15 mM succinic acid/20 mM MgSO$_4$. CtrA antiserum detects several nonspecific bands on the blot, the band corresponding to CtrA was identified by including a cell extract of EA1 (Δ*ctrA*::*aadA*) in the experiment, and is indicated with an arrow.

*C. sphaeroides* would enable strain BV11 (*xrpA*+) to swim in minimal medium containing 15 mM succinic acid/20 mM MgSO$_4$. For these experiments, we used strain BV10 (*fliF1*::*aadA* Δ*osp*::Hyg) instead of BV11 (Δ*fleQ*::Kan Δ*ops*::hyg) to allow for the replacement of the wild-type *rpoN* genes (1 to 4) with the Δ*rpoN*::Kan alleles (1 to 4) [5]. Importantly, we previously verified that the swimming phenotype of the BV10 and BV11 strains was similar (S7 Fig).

Upon deletion of *rpoN3*, BV10 (*fliF1*::*aadA* Δ*osp*::Hyg) was able to swim in minimal medium containing 15 mM succinic acid/20 mM MgCl$_2$, a phenomenon not observed with the deletion of the other three *rpoN* genes (Fig 8A). The introduction of the pRK_rpoN3 plasmid into strain BV23 (BV10, Δ*rpoN3*) was sufficient to restore the BV10 phenotype (Fig 8B). These results suggest that RpoN3 is the σ$^{54}$ factor responsible for *xrpA* expression. In agreement with this notion, XrpA was detected by Western blot in total cell extracts obtained from the Δ*rpoN1*, Δ*rpoN2* and Δ*rpoN4* strains; however, a severe reduction in XrpA was observed in total cell extracts from strain BV23 (Δ*rpoN3*) (Fig 8C).

Eσ$^{54}$ is strictly dependent on an activator protein to initiate transcription. These activator proteins belong to the bacterial enhancer binding protein (bEBP) family, whose members present conserved sequences involved in ATP hydrolysis and interaction with the DNA-Eσ$^{54}$ complex [7–9]. Therefore, we tested whether one of the four bEBPs identified in the *C. sphaeroides* genome with a DNA binding domain [6] could be required for *xrpA* expression. For this purpose, we deleted the *nifA*, *fleQ*, RSWS8N_13400, and RSWS8N_04525 genes in strain BV10. As shown in Fig 9A, only the inactivation of RSWS8N_13400

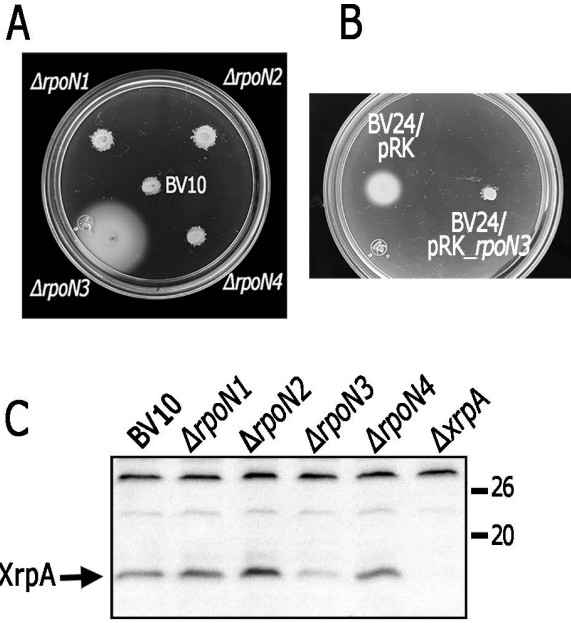

**Fig 8. Detection of XrpA by Western blot and swimming ability of BV10 and its derivative strains generated upon deletion of the different *rpoN* genes.** A, Soft-agar plates were prepared using Sistrom's minimal medium containing 15 mM succinic acid/20 mM $MgSO_4$. Plates were inoculated with strain BV10 and its derivatives carrying the indicated mutation. The plates were incubated heterotrophically for 48 h. B, Soft-agar plates with medium containing 15 mM succinic acid/20 mM $MgSO_4$ were inoculated with BV23 (BV10 derivative carrying Δ*rpoN3*) carrying pRK415 (pRK) or pRK_rpoN3. Plates were incubated heterotrophically for 30 h. C, Detection of XrpA by Western blot in total cell extracts using anti-XrpA antibody. Total cell extracts of BV10 derivatives carrying the indicated mutation were obtained from cultures grown heterotrophically in Sistrom's minimal media containing 15 mM succinic acid/20 mM $MgSO_4$.

enabled the swimming of strain BV10, suggesting that this activator protein, together with RpoN3, is responsible for *xrpA* expression. In agreement with this proposal, XrpA was detected by Western blot in total cell extracts obtained from strains lacking *nifA*, *fleQ*, and RSWS8N_04525; however, a severe reduction was observed in total cell extracts obtained from ΔRSWS8N_13400 (Fig 9B).

The product of RSWS8N_13400, hereafter referred to as AprX (activator protein of *xrpA*), is similar to other canonical bEBPs [50] and is composed of three well-defined domains (Fig 9C). The N-terminal regulatory domain consists of a phosphoacceptor receiver domain (REC domain), similar to other bEBPs, which are controlled by phosphorylation. The central catalytic AAA + domain contains conserved residues involved in ATP binding and hydrolysis, as well as the GAFTGA/GSFTGA motif, which is involved in interactions with $E\sigma^{54}$ [8,51,52]. Finally, the C-terminal region contains the DNA-binding domain.

### *xrpA* expression is modulated by growth conditions

Given that the expression of *xrpA* from the *lac* promoter present in pRK415 leads to a severe reduction in swimming proficiency and flagella formation, even under conditions known to promote swimming (Fig 6C), we hypothesized that *xrpA* expression might be influenced by the growth conditions. To test this possibility, we constructed a transcriptional fusion of the wild type *xrpA* promoter region with the reporter gene *uidA*. The DNA fragment carrying this fusion was cloned into the pRK415 plasmid in the direction opposite to the *lac* promoter present in the vector. The amount of β-glucuronidase was assayed in total cell extracts obtained

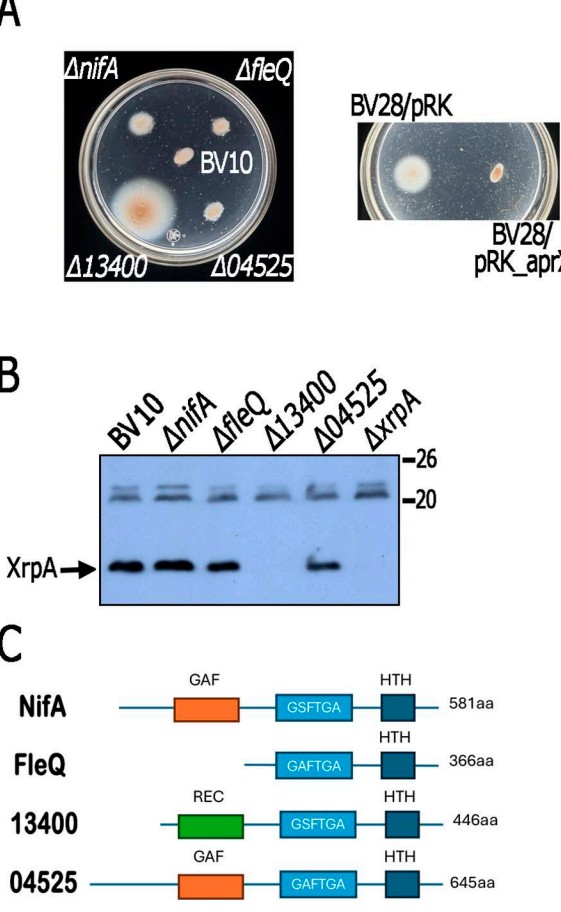

**Fig 9. Detection of XrpA by Western blot and swimming phenotype of BV10 and its derivative strains lacking one of the four genes encoding bona fide bEBPs in *C. sphaeroides*.** A, Soft-agar plates were prepared using minimal media containing 15 mM succinic acid/20 mM MgSO$_4$. Plates were inoculated with strain BV10 and its derivatives carrying the indicated mutation. BV28 is a BV10 derivative carrying Δ*aprX* (ΔRSWS8N_13400). Plates were incubated heterotrophically during 48 h. B, Detection of XrpA by Western blot in total cell extracts using anti-XrpA antibody. Total cell extracts of the indicated strains were obtained from cell cultures grown heterotrophically in Sistrom's minimal medium containing 15 mM succinic/20 mM MgSO$_4$. C, Domain architecture of the 4 bona fide bEBPs containing a DNA binding domain, encoded in *C. sphaeroides* genome [6]; these proteins present the central AAA-ATPase domain, containing the GAFTGA/GSFTGA motif that is involved in the interaction with Eσ$^{54}$, a helix-turn-helix (HTH) domain, and three show an N-terminal regulatory domain.

from strain BV11 carrying this plasmid (pRK_xrpA::uidA). We observed that *xrpA* expression was higher under heterotrophic growth conditions than under photoheterotrophic conditions (Fig 10). Moreover, under heterotrophic growth conditions, *xrpA* expression was higher in 15 mM succinic acid than in 0.2% casamino acids, regardless of the presence of MgSO$_4$ (Fig 10). These results suggest that *xrpA* is transcriptionally controlled by an unknown mechanism that is responsive to specific growth conditions.

## *xrpA* is conserved across different species within the Rhodobacterales order

Genes similar to *xrpA* are found in several species from the *Paracoccaceae* and *Roseobacteraceae* families. This gene is located in a syntenic region upstream of a *hisN* paralog [53] and

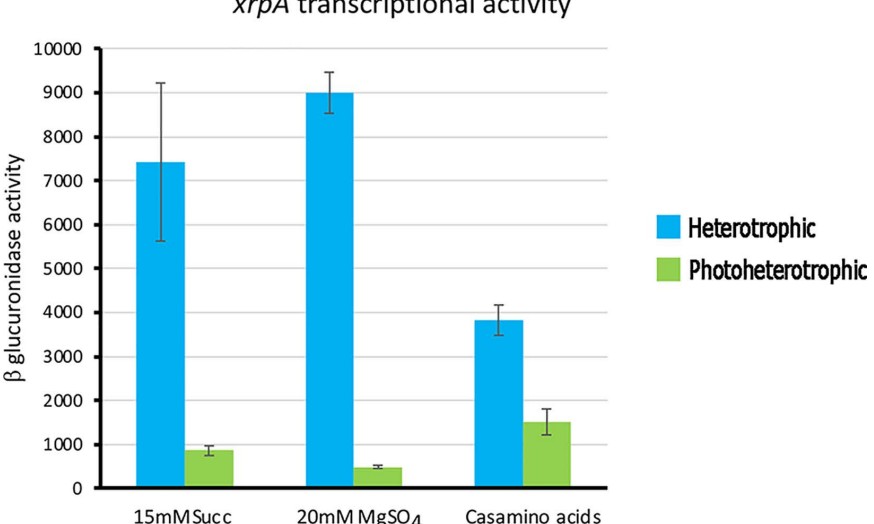

**Fig 10. β-glucuronidase activities promoted by the transcriptional fusion *xrpA::uidA*.** β-glucuronidase activity was measured in total cell extracts obtained from strain BV11/pRK_xrpA::uidA grown under the specified conditions. Activities are expressed as picomoles of methylumbelliferone formed per minute per milligram of protein. The values are the average of three independent experiments, standard deviations are shown.

downstream of a divergently orientated gene encoding an NADPH:quinone oxidoreductase (S8 Fig ). In *C. sphaeroides*, the distance between *xrpA* and *hisN* (89 bp) suggests that *xrpA* may be a monocistron. Further analyses were conducted to investigate whether the presence of *xrpA* cooccurs with homologs of AprX (with a conserved GAFTGA/GSFTGA box) and RpoN. The phylogenetic tree in Fig 11, which includes species in which *xrpA* is found in the abovementioned gene context, revealed that the presence of AprX strongly correlates with the presence of a σ54-dependent promoter upstream of *xrpA*, as well as with the presence of an *rpoN* homolog. This association suggests that the control of *xrpA* by AprX and RpoN is conserved in multiple species, predominantly within *Paracoccaceae* but also within a few species of the *Roseobacteraceae* family. An exception to this correlation was observed for *Phaeovulum veldkampiii*, where no AprX homolog was identified; however, a sequence resembling the σ54 consensus promoter was still detected upstream of *xrpA*. This may indicate that *aprX* was recently lost in this species.

## Discussion

In *C. sphaeroides*, the expression of the TCS CckA/ChpT/CtrA is dispensable, and it remains inactive under standard laboratory growth conditions [13,14]. This situation contrasts with findings in other well-characterized alpha-proteobacteria, where the system is either essential or active under the growth conditions in which these bacteria are routinely grown. This characteristic makes *C. sphaeroides* an interesting model for studying the regulation of this TCS and its role in controlling the flagellar motility system, which is proposed to be an ancestral trait controlled by CtrA [17,54]. In *C. sphaeroides*, characterization of the CckA/ChpT/CtrA system led to the recent discovery of the response regulator Osp, which inhibits CckA autophosphorylation and consequently downregulates the genes controlled by CtrA. The gene encoding Osp is conserved across several species of Rhodobacterales, suggesting that the control of CckA by Osp is a conserved trait in this bacterial group [15].

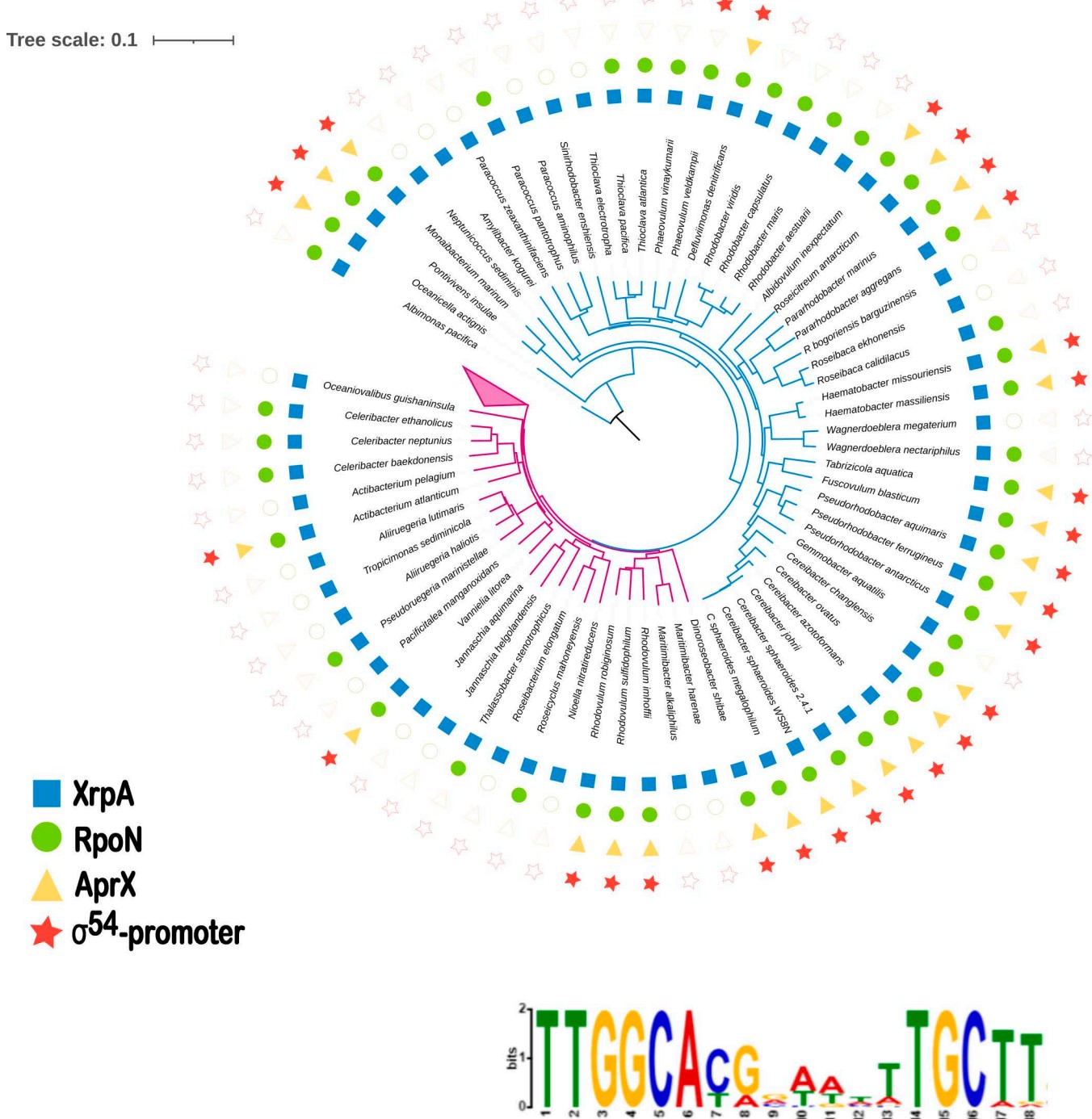

**Fig 11. Co-occurrence of AprX with a σ⁵⁴-promoter consensus sequence upstream of *xrpA* in Rhodobacterales.** A, The species phylogeny is based on RpoC and includes representative species that have *xrpA*. The Family is indicated by the branch color: blue (Paracoccaceae) and magenta (Roseobacteraceae). The presence of RpoN and AprX is indicated by green dots and yellow triangles, respectively. The red stars indicate the presence of a putative σ⁵⁴-promoter upstream of *xrpA*. The collapsed branches include 70 species of Roseobacteraceae that do not have an AprX homologue and do not show a σ⁵⁴-promoter sequence upstream of *xrpA*. Complete information for each species, including the GenBank accession number for each genome, as well as for the proteins RpoC, XrpA, and AprX, is provided in S2 Table. B, Sequence LOGO representation of the conserved motif found upstream of *xrpA*; the sequences from the 28 species that have an *xrpA* homologous gene were included in the analysis. The sequence is highly similar to the RpoN binding motif previously reported for *S. typhimurium* (see Fig 4).

XrpA is one of the few proteins reported thus far that represses the expression of *ctrA*. For example, in *C. crescentus*, SciP and CtrA itself negatively regulate *ctrA* by controlling one of its three promoters [55–58]. However, this regulatory mechanism is tied to the need to modulate CtrA during the cell cycle in this bacterium, which mainly responds to internal cues. In contrast, the control of *ctrA* mediated by XrpA is noticeably influenced by external stimuli, supporting the idea that, in *C. sphaeroides* this system is central to respond to different environments [21]. It is also shown that XrpA represses the expression of *chpT*, which is unusual since, to the best of our knowledge, this gene has not been reported to be part of the CtrA regulon [21,59–61] or to be transcriptionally regulated. Therefore, the negative control of XrpA on *chpT* expression could be relevant in preventing its accumulation in the absence of CtrA, probably avoiding spurious crosstalk with other histidine kinases.

Considering the regulation of *xrpA* by environmental cues, we noticed that the expression of *xrpA* is higher under heterotrophic growth conditions than under photoheterotrophic conditions, suggesting that XrpA may attenuate the CtrA-controlled response during heterotrophic growth in the presence of a preferred carbon source. The rational for this regulation is not obvious but our results suggest that under certain conditions a specific signal could activate CckA (in this case, $MgSO_4$), but the growth conditions through an uncharacterized mechanism would activate the expression of XrpA to counteract CckA activation.

Our findings demonstrate that the expression of *xrpA* is dependent on RpoN3 and the bEBP AprX. Typically, bEBP activity is regulated by one of the following mechanisms: phosphorylation of its N-terminal domain by a cognate histidine kinase, interaction with another protein, or binding of effector molecules [50]. The N-terminal region of AprX contains a canonical REC domain (cd17572) commonly found in bEBPs that are regulated by phosphorylation, suggesting that AprX could be controlled by phosphorylation of this domain. However, given that none of the genes near *aprX* encode a histidine kinase, the identity of this hypothetical protein remains to be discovered. Alternatively, despite having a REC domain, AprX could be controlled by a mechanism other than phosphorylation. For example, several bEBPs are known to be controlled by interaction with another protein, or direct binding of effector a ligand [62–65].

Another important aspect revealed from our results is the discovery of the role of RpoN3, the third paralog among the four *rpoN* genes encoded in the genome of *C. sphaeroides*.

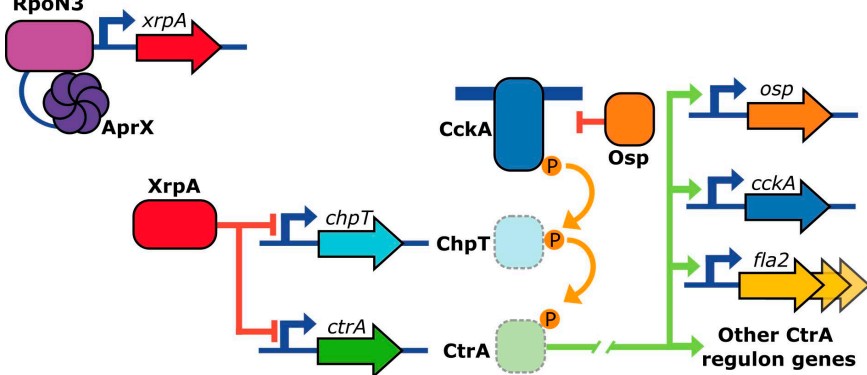

**Fig 12. Regulation of the TCS CckA/ChpT/CtrA in *C. sphaeroides*.** The CtrA regulatory pathway, including the phosphotransfer events, is shown. CckA phosphorylates CtrA via the phosphotransferase ChpT. Osp inhibits the kinase activity of CckA. The RNA polymerase (not represented), associated with the alternative sigma factor RpoN3, and its activator protein AprX, activate the expression of *xrpA*. XrpA, represses the expression of *ctrA* and *chpT*, and, consequently, the expression of the CtrA regulon that includes the *fla2* genes.

*C. sphaeroides* is an exceptional microorganism in which each of the RpoN copies that have been studied thus far, interacts with a particular activator protein (bEBP) to promote the expression of a subset of genes. RpoN1 is known to specifically transcribe the *nif* genes together with its cognate bEBP NifA, while RpoN2 transcribes the *fla1* genes along with its cognate bEBP FleQ. Our findings show that RpoN3 and the bEBP AprX are responsible for the expression of *xrpA*. Further research, now including this pair of proteins, will contribute to a better understanding of the mechanisms underlying the functional specificity among the different RpoN proteins in *C. sphaeroides*. For RpoN1 and RpoN2, the differential recognition of the -11 position in the conserved promoter sequence, along with specific interactions between the DNA-RpoN complex and its cognate bEBP, contribute to transcriptional specificity [5,10].

One intriguing aspect of our findings is the stimulatory effect of $MgSO_4$ on the expression of CtrA-regulated genes in the absence of XrpA. Since $MgSO_4$ does not significantly affect *xrpA* expression, it may influence the CckA/ChpT/CtrA system, either directly or through another regulatory protein. In the marine gamma-proteobacterium *Vibrio fischeri*, the addition of 35 mM $Mg^{2+}$ to the culture medium has a stimulatory effect on motility and flagellation [66]. This effect appears to be associated with a significant reduction in the intracellular levels of c-di-GMP. Among the 50 genes encoding proteins with domains known to be involved in the synthesis and degradation of this second messenger (i.e., GGDEF, EAL and HY-GYP), only six are involved in controlling motility in the presence of $Mg^{2+}$ in this bacterium [67,68]. In *C. crescentus* it has been shown that c-di-GMP inhibits the kinase activity of CckA [69,70]; however, in *C. sphaeroides* no connection between the CckA/ChpT/CtrA system and c-di-GMP has been established yet, but ongoing studies are investigating this potential link.

The inclusion of XrpA in the regulatory network of *C. sphaeroides* is illustrated in Fig 12. It is possible that *xrpA* may serve as a regulatory node where different signals converge to control its expression, thus modulating the CckA/ChpT/CtrA system.

## Supporting information

**S1 Fig. Swimming proficiency of strains AM1, SP13 and BV11 grown under different conditions.** Soft-agar plates containing minimal medium containing the indicated carbon source were inoculated with the indicated strains and incubated either under photoheterotrophic or heterotrophic conditions for 48 h.
(TIF)

**S2 Fig. Effect of different salts on the swimming proficiency of strain AM1.** Soft-agar plates containing minimal medium with 15 mM succinic acid as the carbon source were supplemented with different salts at different concentrations. After inoculation, the plates were incubated under heterotrophic conditions for 48 h. At the top of the figure, the swimming ring formed at the highest concentration of the tested salt is shown. Below, the normalized values of the swimming ring for each salt concentration are shown. Normalization was done by subtracting the measurements taken in the absence of salt. The values determined in the absence of salt were used as the reference values. Measures represent the average of three experiments, each with four replicate swimming plates. The standard deviation is indicated.
(TIF)

**S3 Fig. Swimming proficiency of strains AM1, BV12 (AM1 Δ*osp*::Hyg), SP13 and LC5/ pRK_cckAL391F.** Soft-agar plates containing minimal media with the indicated carbon source were inoculated with the indicated strains and incubated either under heterotrophic or photoheterotrophic conditions for 48 h.
(TIF)

**S4 Fig. Swimming proficiency of SP13 and BV30 (SP13 Δ*xrpA*::Hyg).** Soft-agar plates containing minimal media with 15 mM succinic acid/20 mM MgSO$_4$ were inoculated with the indicated strains and incubated heterotrophically for 48 h.
(TIF)

**S5 Fig. Immunodetection of XrpA in total cell extracts of BV11 and AM1/pRK_xrpA.** Cells were grown heterotrophically in Sistrom's minimal medium containing 15 mM succinic acid, 15 mM succinic acid/20 mM MgSO$_4$, or photoheterotrophically in 0.2% casamino acids.
(TIF)

**S6 Fig. Primary sequence of XrpA and predicted structure.** The secondary structure prediction of XrpA was obtained using Psipred [39]. Blue arrows represent α-helixes. At the left, the ribbon view of the XrpA structure predicted by AlphaFold is shown (AF-Q3J594-F1-v4 for the identical protein Q3J594/RSP_1892 from *C. sphaeroides* 2.4.1) [40, 41].
(TIF)

**S7 Fig. Swimming proficiency of AM1, BV11 and BV10.** Soft-agar plates containing minimal media with the indicated carbon source were inoculated with the indicated strains and incubated under heterotrophic or photoheterotrophic conditions for 48 h.
(TIF)

**S8 Fig. *xrpA* gene context organization in selected α-proteobacteria Synteny was evaluated using the Genomic Context Visualizer (GeCoViz https://gecoviz.cgmlab.org) [42], COG1396 was used as query. In species with several hits, an additional screening based on size, detection of additional domains and genomic neighborhood was carried out. Homologues of *xrpA* are indicated by a vertical black arrow at the top, and the gene is represented by a brown arrow.**
(TIF)

**S1 Table. Oligonucleotides used in this work.**
(DOCX)

**S2 Table. Species and accession numbers of the sequences used to obtain the phylogenetic tree.**
(XLSX)

**S1 Raw Images. File containing all the original blot and gel images included in this work.**
(DOCX)

## Acknowledgments

We thank Aurora Osorio and Luis David Ginez for technical support. Daniel Garzón for his help with antibody production and animal care. We also thank the Molecular Biology Unit and the Microscopy Unit of the Instituto de Fisiología Celular, UNAM for sequencing facilities and support with electron microscopy.

## Author contributions

**Conceptualization:** Benjamín Vega-Baray, Laura Camarena.

**Formal analysis:** Benjamín Vega-Baray, José Hernández-Valle, Sebastián Poggio, Laura Camarena.

**Funding acquisition:** Sebastián Poggio, Laura Camarena.

**Investigation:** Benjamín Vega-Baray, José Hernández-Valle.

**Methodology:** Benjamín Vega-Baray.

**Project administration:** Laura Camarena.

**Supervision:** Sebastián Poggio, Laura Camarena.

**Validation:** José Hernández-Valle.

**Visualization:** Benjamín Vega-Baray, Laura Camarena.

**Writing – original draft:** Benjamín Vega-Baray, Sebastián Poggio, Laura Camarena.

**Writing – review & editing:** Sebastián Poggio, Laura Camarena.

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
