## [Decision Letter · Decision Letter 0]

10 Jan 2025

PONE-D-24-58434Repression of ctrA and chpT by a transcriptional regulator of the Xre family that is expressed by RpoN3 and its cognate activator protein in CereibactersphaeroidesPLOS ONE

Dear Dr. Camarena,

Thank you for submitting your manuscript to PLOS ONE. Two expert reviewers evaluated the merits of the manuscript. As noted below, both reviewers are positive but have a few additional concerns and suggestions that I would like you to address in a revised manuscript.

We look forward to receiving your revised manuscript.

Kind regards,

MD A MOTALEB, Ph.D.

Academic Editor

PLOS ONE

Journal Requirements:

“PAPIIT IN215023 (PAPIIT/DGAPA-UNAM) to LC”

“We thank Aurora Osorio for technical support. Daniel Garzón for his help with antibody production and animal care. We also thank the Molecular Biology Unit and the Microscopy Unit of the Instituto de Fisiología Celular, UNAM for sequencing facilities and support with electron microscopy. This work was partially supported by grant IN215023 (PAPIIT/DGAPA-UNAM) to LC.”

“PAPIIT IN215023 (PAPIIT/DGAPA-UNAM) to LC”

4. Please expand the acronym “PAPIIT/DGAPA-UNAM” (as indicated in your financial disclosure) so that it states the name of your funders in full.

Reviewers' comments:

Reviewer's Responses to Questions

**Comments to the Author**

1. Is the manuscript technically sound, and do the data support the conclusions?

Reviewer #1: Yes

Reviewer #2: Yes

2. Has the statistical analysis been performed appropriately and rigorously? 

Reviewer #1: Yes

Reviewer #2: Yes

3. Have the authors made all data underlying the findings in their manuscript fully available?

Reviewer #1: Yes

Reviewer #2: Yes

4. Is the manuscript presented in an intelligible fashion and written in standard English?

Reviewer #1: Yes

Reviewer #2: Yes

5. Review Comments to the Author

Reviewer #1: The authors focused on the regulation mechanism of flagellar expression in Cereibacter sphaeroides. They found several new signaling components involved in the expression of Fla2 protein and proposed a gene-regulation model for assembling multiple polar flagella. The experiments are well-designed and proceed carefully based on the detailed discussion of the results. I suggest several modifications to the manuscript as follows.

(Figure 2) Seeing the photos under the photoheterotrophic condition, I cannot understand the difference in the expansion of swimming rings between AM1 and SP13; both strains’ rings seem to be expanded similarly by MgSO4. How were the ring diameters measured?

In addition, as in the other figures, please add indications of each panel by A, B, C, and D.

(lines 280-283) The authors state “Conversely, in plates incubated under the photoheterotrophic condition, the inclusion of MgSO₄ only slightly promoted the swimming of BOTH strains ( ~2.5-fold) (Fig. 2).” The right panels of Fig 2 show the swimming diameter of BV11under photoheterotrophic condition is larger than that under heterotrophic condition. It would be better to revise the sentence to “Conversely, in plates incubated under photoheterotrophic conditions, the inclusion of MgSO₄ only slightly promoted the swimming of AM1.” Or, please explain the enhancement of BV11 motility under the photoheterotrophic condition.

(Figure 3) It is difficult to see the scale bars in the EM photos. Please use a larger font size.

(lines 389-390) The authors state “the OVEREXPRESSION of XrpA leads to the inhibition of Fla2-dependent motility even under these conditions, in which BV11 (xrpA+) is able to swim.” Fig. 6A shows that the expression levels of XrpA from BV11 and AM1/pRK_xrpA appear to be comparable. What is the rationale for the overexpression of XrpA in AM1/pRK_xrpA?

(Figure 12) It would be better to include flaA2 in Figure 12, because the control of flaA2 expression is the goal of the presented signaling network.

Reviewer #2: The authors report here on factors that control the expression of one of the two sets of flagellar genes (fla2 genes) in the alpha-proteobacterium Cereibacter sphaeroides. Expression of the fla2 genes is controlled by the CckA/ChpT/CtrA two-component system. The authors identify here a new transcriptional regulator, XrpA, that negatively regulates expression of the fla2 genes by reducing expression of CtrA. They further show that XrpA binds to the promoter regions of ctrA and chpT, and presumably represses transcription of these genes. They also identify one of the four paralogs of the alternative sigma factor RpoN (RpoN3) and its cognate activator (a protein that they designate as AprX) are required for transcription xrpA. The results are well documented and of broad interest. The manuscript is well written.

Specific comments:

1. The authors mention that four bEBPs with a DNA-binding domain have been identified in the C. sphaeroides genome (lines 489-490). This seems to imply that there are bEBPs in C. sphaeroides that lack a DNA-binding domain. Is this correct? If it is true, it would be informative to briefly discuss these bEBPs.

2. Was expression of xrpA examined under nitrogen-fixing or limiting nitrogen conditions? The authors report that under heterotrophic growth conditions, xrpA expression was higher when succinate was included as the carbon source versus casamino acids. When using casamino acids as a carbon source, the bacteria presumably are excreting ammonia since the demand of carbon would be greater than the demand for nitrogen. This might explain the lower expression of xrpA with casamino acids as the carbon source.

3. Bacteria that have two flagellar systems typically elaborate a polar flagellum that is used for swimming and multiple lateral flagella that are used for swarming. That is not the case here, and instead both sets of flagella are used for swimming. It would be helpful if the authors pointed out this distinction.

6. PLOS authors have the option to publish the peer review history of their article (what does this mean? ). If published, this will include your full peer review and any attached files.

**Do you want your identity to be public for this peer review?** For information about this choice, including consent withdrawal, please see our Privacy Policy .

Reviewer #1: No

Reviewer #2: No

---

## [Author Response · Author response to Decision Letter 1]

22 Feb 2025

RESPONSE TO REVIEWERS

Reviewer #1: …..I suggest several modifications to the manuscript as follows.

(Figure 2) Seeing the photos under the photoheterotrophic condition, I cannot understand the difference in the expansion of swimming rings between AM1 and SP13; both strains’ rings seem to be expanded similarly by MgSO4. How were the ring diameters measured?

In addition, as in the other figures, please add indications of each panel by A, B, C, and D.

(Figure 2) Swimming plates are inoculated with a 2 µl drop of each strain, if the drop slides on the surface of the culture medium, we observe a lopsided denser growth zone and if the strain can swim a less dense swimming ring appears. In Fig 2C in the presence of MgSO4, SP13 only shows the lopsided growth, while the AM1 strain exhibits the lopsided growth and the swimming ring. When this occurs, we measure the shortest diameter of the ring. This information was included in Material and Methods (lines 160-163).

As suggested, we have added indications of each panel by A, B, C and D.

(lines 280-283) The authors state “Conversely, in plates incubated under the photoheterotrophic condition, the inclusion of MgSO₄ only slightly promoted the swimming of BOTH strains ( ~2.5-fold) (Fig. 2).” The right panels of Fig 2 show the swimming diameter of BV11under photoheterotrophic condition is larger than that under heterotrophic condition. It would be better to revise the sentence to “Conversely, in plates incubated under photoheterotrophic conditions, the inclusion of MgSO₄ only slightly promoted the swimming of AM1.” Or, please explain the enhancement of BV11 motility under the photoheterotrophic condition.

The purpose of Figure 2A is to explain why heterotrophic growth conditions utilizing 15 mM succinic acid as a carbon source were chosen for the subsequent experiments in this work. As it can be seen, the difference in the response of the AM1 and BV11 strains to the presence of MgSO4 is stronger under this condition, making it more suitable to study the effect of XrpA. We have slightly modified the original text to clarify this point (lines 290-293).

(Figure 3) It is difficult to see the scale bars in the EM photos. Please use a larger font size.

(Figure 3) As suggested, we have increased the font size and the scale bar.

(lines 389-390) The authors state “the OVEREXPRESSION of XrpA leads to the inhibition of Fla2-dependent motility even under these conditions, in which BV11 (xrpA+) is able to swim.” Fig. 6A shows that the expression levels of XrpA from BV11 and AM1/pRK_xrpA appear to be comparable. What is the rationale for the overexpression of XrpA in AM1/pRK_xrpA?

We have modified the text to indicate that XrpA is expressed in an unregulated manner from the pRK_xrpA plasmid (not overexpressed), and this significantly reduces the swimming ability of AM1/pRK_xrpA when grown in casamino acids (line 399). To further support this idea, we have included a supplementary figure (Fig. S5) to show the differences in XrpA levels under different growth conditions for the BV11 and AM1/pRK_xrpA strains (lines 401-405).

(Figure 12) It would be better to include flaA2 in Figure 12, because the control of flaA2 expression is the goal of the presented signaling network.

(Figure 12) We have included fla2 in Figure 12 as suggested.

Reviewer #2

Specific comments:

1. The authors mention that four bEBPs with a DNA-binding domain have been identified in the C. sphaeroides genome (lines 489-490). This seems to imply that there are bEBPs in C. sphaeroides that lack a DNA-binding domain. Is this correct? If it is true, it would be informative to briefly discuss these bEBPs.

1. C. sphaeroides has five EBPs with the conserved motif GA(F/S)TGA which is necessary to interact with RpoN. The EBP FleT lacks the DNA-binding domain, and its transcriptional action is exerted through FleQ, another EBP. As suggested by the reviewer, we have included information about FleT in the Introduction section (lines 79-84). We believe that this information fits well in this section and clarifies the role of this protein.

2. Was expression of xrpA examined under nitrogen-fixing or limiting nitrogen conditions? The authors report that under heterotrophic growth conditions, xrpA expression was higher when succinate was included as the carbon source versus casamino acids. When using casamino acids as a carbon source, the bacteria presumably are excreting ammonia since the demand of carbon would be greater than the demand for nitrogen. This might explain the lower expression of xrpA with casamino acids as the carbon source.

2. The expression of xrpA has not been examined under nitrogen-fixing or nitrogen-limiting conditions, and the reviewer's suggestion is plausible in explaining the effect of casamino acids. We explored this matter by using the usual amount of (NH4)2SO4 in the culture medium, which corresponds to 3.7 mM, and reducing it to 1.85 mM, 0.74 mM, and 0.37 mM. Total cell extracts of BV11 grown under these conditions did not reveal any notable differences in XrpA levels between the samples. This figure has been uploaded for review.

3. Bacteria that have two flagellar systems typically elaborate a polar flagellum that is used for swimming and multiple lateral flagella that are used for swarming. That is not the case here, and instead both sets of flagella are used for swimming. It would be helpful if the authors pointed out this distinction.

3. We have included this information as suggested (lines 93-95).

---

## [Decision Letter · Decision Letter 1]

4 Mar 2025

Repression of ctrA and chpT by a transcriptional regulator of the Xre family that is expressed by RpoN3 and its cognate activator protein in Cereibactersphaeroides

PONE-D-24-58434R1

Dear Dr. Camarena,

We’re pleased to inform you that your manuscript has been judged scientifically suitable for publication and will be formally accepted for publication once it meets all outstanding technical requirements.

Kind regards,

MD A MOTALEB, Ph.D.

Academic Editor

PLOS ONE

Additional Editor Comments (optional):

Reviewers' comments:

Reviewer's Responses to Questions

**Comments to the Author**

1. If the authors have adequately addressed your comments raised in a previous round of review and you feel that this manuscript is now acceptable for publication, you may indicate that here to bypass the “Comments to the Author” section, enter your conflict of interest statement in the “Confidential to Editor” section, and submit your "Accept" recommendation.

Reviewer #1: All comments have been addressed

Reviewer #2: All comments have been addressed

2. Is the manuscript technically sound, and do the data support the conclusions?

Reviewer #1: Yes

Reviewer #2: Yes

3. Has the statistical analysis been performed appropriately and rigorously? 

Reviewer #1: Yes

Reviewer #2: Yes

4. Have the authors made all data underlying the findings in their manuscript fully available?

Reviewer #1: Yes

Reviewer #2: Yes

5. Is the manuscript presented in an intelligible fashion and written in standard English?

Reviewer #1: Yes

Reviewer #2: Yes

6. Review Comments to the Author

Reviewer #1: (No Response)

Reviewer #2: (No Response)

7. PLOS authors have the option to publish the peer review history of their article (what does this mean? ). If published, this will include your full peer review and any attached files.

**Do you want your identity to be public for this peer review?** For information about this choice, including consent withdrawal, please see our Privacy Policy .

Reviewer #1: No

Reviewer #2: No

---

## [Editor Report · Acceptance letter]

PONE-D-24-58434R1

PLOS ONE

Dear Dr. Camarena,

I'm pleased to inform you that your manuscript has been deemed suitable for publication in PLOS ONE. Congratulations! Your manuscript is now being handed over to our production team.

Kind regards,

on behalf of

Dr. MD A MOTALEB

Academic Editor

PLOS ONE